# Relationships among the β3-adrenargic receptor gene Trp64Arg polymorphism, hypertension, and insulin resistance in a Japanese population

Youhei Yamada[1,2]*, Haruki Nakamura[1], Hiromasa Tsujiguchi[1,2,3], Akinori Hara[1,2,3], Sakae Miyagi[4], Takayuki Kannon[3,5], Takehiro Sato[3,5], Kazuyoshi Hosomichi[3,5], Thao Thi Thu Nguyen[6], Yasuhiro Kambayashi[7], Yukari Shimizu[8], Kim Oanh Pham[1], Keita Suzuki[1,2], Fumihiko Suzuki[1], Tomoko Kasahara[1,2], Hirohito Tsuboi[9], Atsushi Tajima[3,5], Hiroyuki Nakamura[1,2,3]

1 Department of Environmental and Preventive Medicine, Graduate School of Medical Science, Kanazawa University, Kanazawa, Japan, 2 Department of Public Health, Kanazawa University Graduate School of Advanced Preventive Medical Sciences, Kanazawa, Japan, 3 Advanced Preventive Medical Sciences Research Center, Kanazawa University, Kanazawa, Japan, 4 Innovative Clinical Research Center, Kanazawa University, Kanazawa, Japan, 5 Department of Bioinformatics and Genomics, Graduate School of Advanced Preventive Medical Sciences, Kanazawa University, Kanazawa, Japan, 6 Faculty of Public Health, Haiphong University of Medicine and Pharmacy, Haiphong, Vietnam, 7 Department of Public Health, Faculty of Veterinary Medicine, Okayama University of Science, Kanazawa, Japan, 8 Faculty of Health Sciences, Department of Nursing, Komatsu University, Kanazawa, Japan, 9 Institute of Medical, Pharmaceutical, and Health Sciences, Kanazawa University, Kanazawa, Japan

* yamada503597@stu.kanazawa-u.ac.jp

**Data Availability Statement:** All relevant data are within the manuscript.

## Abstract

A polymorphism in the ADRB3 gene (Trp64Arg) has been associated with obesity, insulin resistance, and hypertension. This cross-sectional study investigated the relationships among this polymorphism, hypertension, and insulin resistance values (HOMA-IR) in 719 Japanese subjects aged 40 years and older. The genotype frequencies of Trp64Trp (homozygous, wild), Trp64Arg (heterozygous, variant), and Arg64Arg (homozygous, variant) were 466 (65%), 233 (32%), and 20 (3%), respectively. Insulin resistance was associated with an increased risk of hypertension in a Japanese population. This relationship was dependent on the presence or absence of the Trp64Arg polymorphism (odds ratio, 2.054; confidence interval, 1.191 to 3.541; P value, 0.010). Therefore, the Trp64Arg polymorphism of ADRB3 was associated with hypertension and insulin resistance in a healthy Japanese population. This relationship, which was dependent on the polymorphism, may predict the development of hypertension and diabetes.

## Introduction

Hypertension is a major risk factor for global disease burden [1]. Many patients with hypertension have diabetes mellitus, which is strongly related to coronary heart disease, major stroke

**Funding:** The authors received no specific funding for this work.

**Competing interests:** The authors have declared that no competing interests exist.

subtypes, and deaths attributed to other vascular causes [2]. The pathophysiology of these two diseases are similar and related to obesity and insulin resistance. Insulin resistance is a pathological condition that impairs insulin sensitivity. Previous studies reported a close relationship between hypertension and insulin resistance [3].

The etiology of hypertension involves genetic disorders. The present study on hypertension was based on Genome-wide Association Studies (GWAS), which search a vast number of single nucleotide polymorphisms (SNP) in a large cohort [4]. Prior to the initiation of GWAS, the main strategy employed to identify hypertension-susceptibility genes was the candidate gene approach. Several candidate genes have been reported using conventional approaches [5].

The β3-adrenergic receptor (ADRB3), one of the candidate genes, is a class of G protein-coupled receptors that primarily mediates lipolysis and thermogenesis. A polymorphism in the ADRB3 gene (Trp64Arg) impairs the function of ADRB3. A decline in function causes the pathogenesis of multiple conditions, including hypertension, insulin resistance, and obesity [6–8].

The effects of the Trp64Arg polymorphism need to be considered in investigations on the relationship between hypertension and insulin resistance. However, few studies have examined this triangular relationship. Widén et al. reported that the ratio of hypertension and insulin resistance was higher in Finns with the Trp64Arg polymorphism [7]. However, Fujisawa et al. suggested that the Trp64Arg polymorphism did not markedly affect the development of hypertension or insulin resistance in Japanese individuals [9].

Therefore, the present study aimed to examine the relationships among hypertension, insulin resistance, and the Trp64Arg polymorphism.

## Methods

### Study design and participants

Comprehensive medical check-up data obtained from the residents of Shika town, a rural area in Japan, were used for the analysis. Baseline data were derived from the SHIKA study, an overview of which was previously reported [10]. In brief, the SHIKA study is a population-based observational study conducted to investigate approaches that prevent lifestyle-related diseases. It was conducted with the approval of the Ethics Committee of Kanazawa University and informed consent was obtained from all participants. The target subjects of the SHIKA study were all middle-aged residents who were delivered a self-administrated questionnaire and requested to undergo a comprehensive health examination. In the present study, data on 1191 voluntary participants from 40 years of age who underwent the comprehensive health examination between March 2014 and January 2017 were available. The design of the present study was cross-sectional. This study was conducted with the approval of the Ethics Committee of Kanazawa University. Written informed consent was obtained from all participants.

Subjects with incomplete data on SNP (n = 326), blood pressure (n = 5), or fasting blood sugar (n = 83) and those whose HOMA-IR (homeostasis model assessment for insulin resistance) was less 0.3 (n = 18) were excluded from the analysis. Therefore, this study ultimately included 719 subjects (Fig 1).

### Genotyping

Genomic DNA was extracted from blood samples using the QIAamp DNA Blood Maxi Kit (QIAGEN Inc., Venlo, Netherlands) according to the manufacturer's instructions or consigned to a company specialized in clinical laboratory testing (SRL, Inc., Tokyo, Japan). SNP genotyping was performed using the Japonica Array v2 [11] (TOSHIBA Inc., Tokyo, Japan).

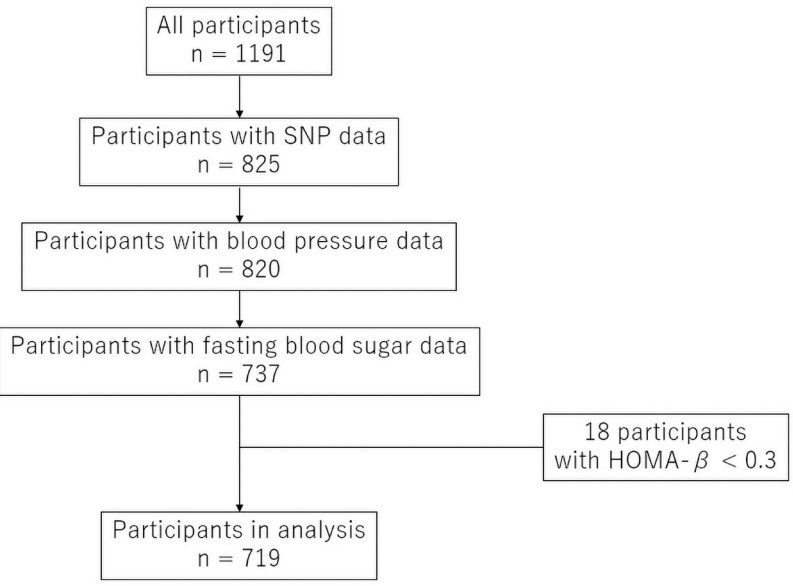

**Fig 1. Participant recruitment chart.**

The genotypes of ADRB3 Trp64Arg (rs4994) in 825 unrelated subjects (based on genome-wide $\hat{\pi}$ values) were extracted from array data. The call rate for SNP was 100% and a departure from the Hardy-Weinberg equilibrium was not observed.

## Blood pressure measurement

Well-trained nurses and clinical technologists measured blood pressure (BP) using a fixed protocol. Two automated digital sphygmomanometers, HEM-907 (OMRON Inc, Kyoto, Japan) and UM-15P (Parama-tech Inc., Fukuoka, Japan), were used to check BP and their measurement of principle, the oscillometric method, was the same. This medical check-up was conducted in the morning and BP was measured in a fasted state.

BP was measured twice consecutively in a sitting position with an appropriate cuff and averages were adopted as BP data.

Subjects were divided into two groups according to the following definition of hypertension: subjects diagnosed with hypertension and being treated with antihypertensive drugs or those with BP of higher than 140/90 mmHg in medical check-ups.

## Assessment of insulin resistance

HOMA-IR is regarded as a robust index for the assessment of insulin resistance and the homeostasis model assessment of beta cell function (HOMA-β) has been proven as a reliable tool for the assessment of insulin secretion. These indexes are widely used in large population studies [12]. HOMA-IR and HOMA-β are assessed using the following equations: HOMA-IR = (FPI × FPG) / 405, HOMA-β = (360 × FPI) / (FPG − 63). FPI is an aberration of fasting plasma insulin concentration (μIU/mL) and FPG is that of fasting plasma glucose (mg/dL).

## Other variables

Daily habits were assessed using self-administered questionnaires. Subjects who had a habit of exercising for more than 30 minutes at least twice a week for 1 year or habitually

performed tasks such as carrying baggage, walking, and cleaning for more than 1 hour a day were regarded as subjects with an exercise habit. The frequency of drinking was classified into two groups according to answers to the following questions: "Do you drink more than one glass of sake (22 g ethanol) per day three times a week?" or "Do you drink at least four times a week?". A drinking habit was confirmed by replying in the affirmative to either of these questions.

## Statistical analysis

The Student's *t*-test was used to compare the average of continuous variables and the chi-squared test to compare the proportions of categorical variables. All subjects were stratified into two groups: the BP groups (Hypertension group and Normal BP group) and ADRB3 polymorphism groups (Trp64Trp group and Trp64Arg or Arg64Arg group). A two-way analysis of variance (two-way ANOVA) was used to examine differences in HOMA-IR between the BP groups and ADRB3 polymorphism groups. A multiple logistic regression analysis after adjustments for independent factors was performed to assess the relationship between BP and HOMA-IR.

In all analyses, the threshold for significance was P<0.05. All statistical analysis were performed using IBM SPSS Statistics version 24.0 for Mac (SPSS Inc., Armonk, NY, USA).

## Results

The demographic characteristics of subjects stratified by the genotype of the ADRB3 polymorphism were shown in Table 1.

Genotyping in 719 subjects showed that 466 (65%) were homozygous for the wild-type allele (Trp64Trp), 233 (32%) were heterozygous for the variant allele (Trp64Arg), and 20 (3%) were homozygous for the variant allele (Arg64Arg) (Fig 2).

Among all subjects, 327 were men and 392 were women with a mean age of 61.8 years. Mean SBP and DBP in 368 subjects treated with antihypertensive drugs were 138 and 80.1 mmHg, respectively. Fifty-four subjects were being treated for diabetes and mean HbA1c (NGSP) and HOMA-IR were 5.93% and 1.36, respectively.

Table 2 shows the demographic characteristics of the two ADRB3 polymorphism groups (Trp64Trp group and Trp64Arg or Arg64Arg group), which were similar. No significant differences were observed in age, the prevalence of hypertensive subjects, the use of antihypertensive drugs, SBP, or DBP in each stratified group. Characteristics regarding sugar metabolism (FBS, HbA1c, fasting insulin, HOMA-β, and HOMA-IR) did not significantly differ between the two groups.

When subjects were classified into two groups according to the definition of hypertension, the Hypertension group was significantly older (P < 0.001) and had a higher BMI (P < 0.001) and lower eGFR (P = 0.011) than the Normal BP group (Table 3). The Hypertension group also showed significantly higher FBS (P < 0.001), HbA1c (P = 0.008), fasting insulin (P < 0.001), and HOMA-IR (P < 0.001) than the Normal BP group.

We also assessed differences in variables related to diabetes between the BP groups and ADRB3 polymorphism groups using a two-way ANOVA (Table 4). The results obtained revealed a significant interaction between BP groups and ADRB3 polymorphism groups for HOMA-IR (P = 0.046).

To adjust for the effects of confounding factors, a multiple logistic regression analysis was used to evaluate the relationship between HOMA-IR and hypertension. Based on the significant interaction between the BP groups and ADRB3 polymorphism groups for HOMA-IR

**Table 1. Subject characteristics in three different allele groups.**

| Characteristics | All subjects | Trp64Trp | Trp64Arg | Arg64Arg |
|---|---|---|---|---|
| No. of subjects, n (%) | 719 (100) | 466 (65) | 233 (32) | 20 (3) |
| Men, n (%) | 327 (45) | 203 (44) | 115 (49) | 9 (45) |
| Age | 61.8 (10.7) | 61.5 (10.6) | 62.7 (11.0) | 58.6 (10.1) |
| Hypertensive subjects, n (%) | 368 (51) | 236 (51) | 127 (55) | 5 (25) |
| Use of antihypertensive drugs, n (%) | 238 (33) | 154 (33) | 81 (35) | 3 (15) |
| Use of diabetes treatment, n (%) | 54 (8) | 30 (6) | 23 (10) | 1 (5) |
| Smoking status, n (%) | | | | |
| Non- or ex-smoker | 593 (82) | 384 (82) | 194 (83) | 15 (75) |
| Current | 126 (18) | 82 (18) | 39 (17) | 5 (25) |
| Exercise habit, n (%) | | | | |
| yes | 425 (59) | 277 (59) | 135 (58) | 13 (65) |
| no | 294 (41) | 189 (41) | 98 (42) | 7 (35) |
| Drinking habit, n (%) | | | | |
| yes | 272 (38) | 182 (39) | 81 (35) | 9 (45) |
| no | 447 (62) | 284 (61) | 152 (65) | 11 (55) |
| Height (cm) | 160 (9.10) | 160 (9.06) | 160 (9.32) | 161 (7.19) |
| Weight (kg) | 60.2 (11.8) | 60.1 (11.8) | 60.4 (12.1) | 60.0 (10.0) |
| Waist circumference (cm) | 84.3 (9.10) | 84.3 (9.09) | 84.3 (9.16) | 83.8 (9.31) |
| BMI (kg/m$^2$) | 23.4 (3.21) | 23.5 (3.18) | 23.4 (3.33) | 22.9 (2.74) |
| underweight, n (BMI<18.5) (%) | 33 (5) | 20 (4) | 13 (6) | 0 (0) |
| normal weight, n (18.5≦BMI<25) (%) | 484 (67) | 315 (68) | 151 (65) | 18 (90) |
| overweight, n (BMI≦25) (%) | 202 (28) | 131 (28) | 69 (30) | 2 (10) |
| SBP (mmHg) | 138 (19.1) | 139 (19.7) | 138 (18.1) | 132 (13.6) |
| DBP (mmHg) | 80.1 (11.4) | 80.3 (11.4) | 80.0 (11.7) | 79.5 (8.51) |
| FBS (mg/dL) | 97.0 (18.1) | 96.8 (17.1) | 97.4 (19.7) | 96.9 (22.2) |
| HbA1c (NGSP) (%) | 5.93 (0.649) | 5.92 (0.578) | 5.96 (0.749) | 5.96 (0.925) |
| Insulin (μIU/mL) | 5.52 (3.64) | 5.51 (3.52) | 5.53 (3.90) | 5.84 (3.54) |
| HOMA-β (%) | 65.4 (43.3) | 65.8 (43.3) | 64.3 (44.0) | 69.9 (37.2) |
| HOMA-IR | 1.36 (1.06) | 1.34 (0.959) | 1.38 (1.24) | 1.43 (0.967) |
| eGFR (mL/min/1.73m$^2$) | 72.2 (15.0) | 72.6 (15.1) | 71.4 (14.8) | 73.6 (15.2) |
| Total Cholesterol (mg/dL) | 216 (36.1) | 217 (37.9) | 213 (30.3) | 236 (48.6) |
| LDL Cholesterol (Friedewald) (mg/dL) | 127 (34.0) | 129 (35.2) | 123 (29.4) | 143 (46.9) |
| triglyceride (mg/dL) | 118 (74.3) | 115 (65.9) | 124 (88.8) | 125 (72.6) |

Continuous variables are shown as means (standard deviations). Abbreviations: BMI, body mass index; SBP, systolic blood pressure; DBP, diastolic blood pressure; FBS, fasting blood sugar; HbA1c (NGSP); hemoglobin A1c (National Glycohemoglobin Standardization Program); HOMA-β, homeostatic model assessment beta cell function; HOMA-IR; homeostasis model assessment insulin resistance; eGFR, estimated glomerular filtration rate; LDL, low density lipoprotein.

(Table 4), we performed separate multiple logistic regression analyses according to the presence or absence of the ADRB3 polymorphism (Table 5). In subjects with the ARDB3 polymorphism (Trp64Arg or Arg64Arg), HOMA-IR was inversely associated with hypertension after adjustments for the following confounding factors: sex, age, BMI (Model 1), eGFR, exercise habit, smoking status, drinking habit, treatment of diabetes (Model 2), and HOMA-β (Model 3). On the other hand, no correlation was observed between HOMA-IR and hypertension in those homozygous for the wild-type allele (Trp64Trp) group. These relationships were the same in other multiple regression analysis models.

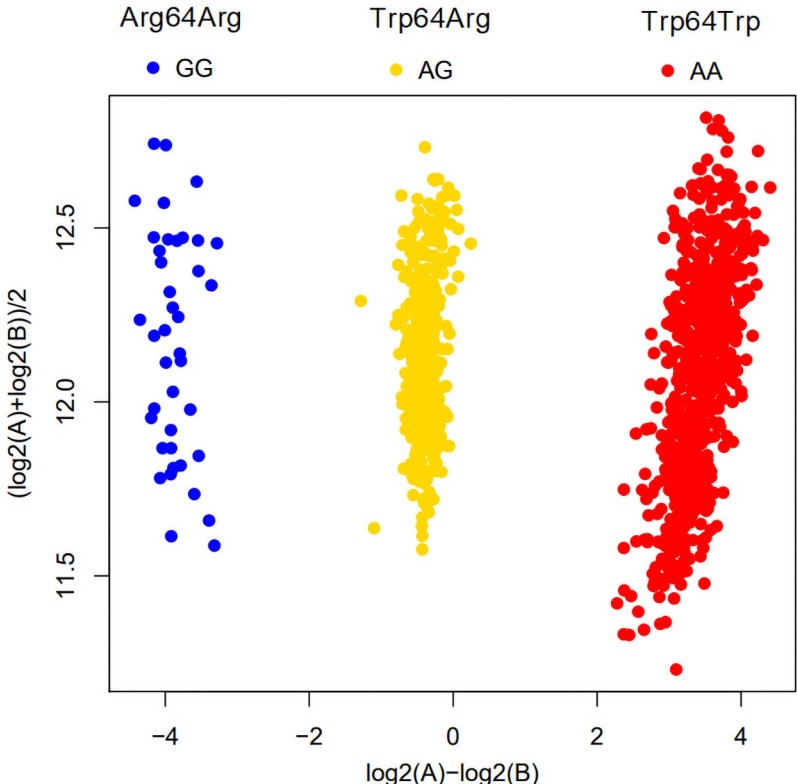

**Fig 2. The genotyping result of rs4994 using the Japonica Array v2. X- and Y-axes indicate logarithmic ratio and logarithmic mean of A and B signals, respectively.**

## Discussion

The present study was conducted in an attempt to examine the relationships among hypertension, insulin resistance, and the Trp64Arg polymorphism. The results obtained suggested that the Trp64Arg polymorphism of ADRB3 was associated with hypertension and insulin resistance.

In the present study, insulin resistance assessed by HOMA-IR was associated with an increased risk of hypertension in a Japanese population. This result is consistent with previous findings showing the important role of insulin resistance in predicting the future incidence of hypertension in middle-aged Japanese men [13]. In this 7-year follow-up study, subjects with the highest baseline insulin resistance values were more likely to become hypertensive after 7 years. This study adds further evidence to the finding that insulin resistance is a risk of hypertension in Japanese.

The present results also revealed that this relationship was dependent on the presence or absence of the Trp64Arg polymorphism. The relationship between high HOMA-IR and a low risk of hypertension appeared to be stronger in subjects who were heterozygous for the variant allele (Trp64Arg) or homozygous for the variant allele (Arg64Arg) than in those who were homozygous for the wild-type allele (Trp64Trp). This correlation was still observed after adjustments for confounding factors.

Previous studies examined the relationships among the ADRB3 polymorphism, obesity [6–8], insulin resistance [6, 7], and hypertension [7]. Walston et al. reported that Pima subjects homozygous for the ADRB3 polymorphism showed the earlier onset of non-insulin dependent

**Table 2. Characteristics of subjects in two different allele groups.**

| Characteristics | Trp64Trp | Trp64Arg or Arg64Arg | P value |
|---|---|---|---|
| No. of subjects, n (%) | 466 (65) | 253 (35) | |
| Men, n (%) | 203 (44) | 124 (49) | 0.161 |
| Age | 61.5 (10.6) | 62.4 (11.0) | 0.291 |
| Hypertensive subjects, n (%) | 236 (51) | 132 (52) | 0.695 |
| Use of antihypertensive drugs, n (%) | 154 (33) | 84 (33) | 0.966 |
| Use of diabetes treatment, n (%) | 30 (6) | 24 (10) | 0.139 |
| Smoking status, n (%) | | | 0.945 |
| Non- or ex-smoker | 384 (82) | 209 (83) | |
| Current | 82 (18) | 44 (17) | |
| Exercise habit, n (%) | | | 0.806 |
| yes | 277 (59) | 148 (58) | |
| no | 189 (41) | 105 (42) | |
| Drinking habit, n (%) | | | 0.358 |
| yes | 182 (39) | 90 (36) | |
| no | 284 (61) | 163 (64) | |
| Height (cm) | 160 (9.06) | 160 (9.16) | 0.423 |
| Weight (kg) | 60.1 (11.8) | 60.4 (12.0) | 0.747 |
| Waist circumference (cm) | 84.3 (9.09) | 84.3 (9.15) | 0.984 |
| BMI (kg/m$^2$) | 23.5 (3.18) | 23.4 (3.29) | 0.826 |
| SBP (mmHg) | 139 (19.7) | 137 (17.9) | 0.338 |
| DBP (mmHg) | 80.3 (11.4) | 79.7 (11.4) | 0.496 |
| FBS (mg/dL) | 96.8 (17.1) | 97.4 (19.9) | 0.660 |
| HbA1c (NGSP) (%) | 5.92 (0.578) | 5.96 (0.763) | 0.478 |
| Insulin (μIU/mL) | 5.51 (3.52) | 5.55 (3.86) | 0.896 |
| HOMA-β (%) | 65.8 (43.3) | 64.7 (43.5) | 0.754 |
| HOMA-IR | 1.34 (0.959) | 1.38 (1.22) | 0.610 |
| eGFR (mL/min/1.73m$^2$) | 72.6 (15.1) | 71.5 (14.8) | 0.371 |
| Total Cholesterol (mg/dL) | 217 (37.9) | 214 (32.6) | 0.401 |
| LDL Cholesterol (Friedewald) (mg/dL) | 129 (35.2) | 125 (31.5) | 0.115 |
| triglyceride (mg/dL) | 115 (65.9) | 124 (87.5) | 0.117 |

Continuous variables are shown as means (standard deviations). p values were from the Student's *t*-test for continuous variables and the chi-squared test for categorical variables. Abbreviations: BMI, body mass index; SBP, systolic blood pressure; DBP, diastolic blood pressure; FBS, fasting blood sugar; HbA1c (NGSP); hemoglobin A1c (National Glycohemoglobin Standardization Program); HOMA-β, homeostatic model assessment beta cell function; HOMA-IR; homeostasis model assessment insulin resistance; eGFR, estimated glomerular filtration rate; LDL, low density lipoprotein.

diabetes mellitus and had a slightly lower resting metabolic rate [6]. Widén et al. suggested that the ADRB3 polymorphism was associated with insulin resistance syndrome, which includes obesity and hypertension, in Finns [7]. Clément et al. showed that individuals with the ADRB3 polymorphism may have an increased capacity to gain weight in France [8]. In contrast to these findings, no significant relationship was observed between the ADRB3 polymorphism and these phenotypes in the present study (Table 2).

The reason for this difference currently remains unclear; however, two possibilities need to be considered. Due to the relatively weak contribution of the ADRB3 polymorphism to insulin resistance and hypertension, the sample sizes of previous studies may have been insufficient. Furthermore, differences in the genetic background may have led to insulin resistance and

**Table 3. Characteristics of subjects in different BP groups.**

| Characteristics | Hypertension | Normal BP | P value |
|---|---|---|---|
| No. of subjects, n (%) | 368 (51) | 351 (49) | |
| Men, n (%) | 181 (49) | 146 (42) | 0.041 |
| Age | 64.5 (10.0) | 59.0 (10.7) | <0.001 |
| ADRB3 (rs4994) | | | 0.695 |
| Trp64Trp | 236 (64) | 230 (66) | |
| Trp64Arg or Arg64Arg | 132 (36) | 121 (34) | |
| Use of antihypertensive drugs, n (%) | 238 (65) | 0 (0) | <0.001 |
| Use of diabetes treatment, n (%) | 33 (9) | 21 (6) | 0.129 |
| Smoking status, n (%) | | | 0.923 |
| Non- or ex-smoker | 304 (83) | 289 (82) | |
| Current | 64 (17) | 62 (18) | |
| Exercise habit, n (%) | | | 0.497 |
| yes | 222 (60) | 203 (58) | |
| no | 146 (40) | 148 (42) | |
| Drinking habit, n (%) | | | 0.006 |
| yes | 157 (43) | 115 (33) | |
| no | 211 (57) | 236 (67) | |
| Height (cm) | 159 (9.29) | 160 (8.89) | 0.302 |
| Weight (kg) | 62.0 (12.2) | 58.3 (11.2) | <0.001 |
| Waist circumference (cm) | 86.3 (9.13) | 82.1 (8.57) | <0.001 |
| BMI (kg/m$^2$) | 24.3 (3.31) | 22.6 (2.89) | <0.001 |
| SBP (mmHg) | 149 (17.3) | 127 (13.8) | <0.001 |
| DBP (mmHg) | 84.7 (12.2) | 75.3 (8.23) | <0.001 |
| FBS (mg/dL) | 100 (17.6) | 94.3 (18.3) | <0.001 |
| HbA1c (NGSP) (%) | 6.00 (0.626) | 5.87 (0.667) | 0.008 |
| Insulin (μIU/mL) | 6.18 (4.11) | 4.84 (2.93) | <0.001 |
| HOMA-β (%) | 67.9 (47.7) | 62.8 (38.1) | 0.113 |
| HOMA-IR | 1.60 (1.22) | 1.15 (0.804) | <0.001 |
| eGFR (mL/min/1.73m$^2$) | 73.7 (15.7) | 73.7 (14.1) | 0.011 |
| Total Cholesterol (mg/dL) | 215 (36.3) | 217 (36.0) | 0.359 |
| LDL Cholesterol (Friedewald) (mg/dL) | 126 (35.0) | 129 (33.0) | 0.280 |
| Triglyceride (mg/dL) | 125 (80.3) | 112 (66.9) | 0.021 |

Continuous variables are shown as means (standard deviations). p values were from the Student's *t*-test for continuous variables and the chi-squared test for categorical variables. Hypertension was defined as the use of antihypertensive medication or a blood pressure of 140/90 mmHg or higher. Abbreviations: BP, blood pressure; BMI, body mass index; SBP, systolic blood pressure; DBP, diastolic blood pressure; FBS, fasting blood sugar; HbA1c (NGSP); hemoglobin A1c (National Glycohemoglobin Standardization Program); HOMA-β, homeostatic model assessment beta cell function; HOMA-IR; homeostasis model assessment insulin resistance; eGFR, estimated glomerular filtration rate; LDL, low density lipoprotein.

hypertension. Another candidate gene to these pathologies may have affected the findings obtained in these studies.

In addition to the ADRB3 gene, genes related to insulin resistance and obesity have been reported in Caucasians. Among all, polymorphisms in the adiponectin gene have been reported to reduce adiponectin levels in overweight and obese children in Italy and increase insulin resistance [14]. Moreover, two Japanese studies reported that polymorphism in the adiponectin gene are associated with insulin resistance [15, 16]. Although we could not obtain

**Table 4. Interaction between BP groups and the ADRB3 polymorphism at rs4994.**

| Variable | ADRB3 polymorphism at rs4994 | Hypertension<br>Average (SD) | Normal BP<br>Average (SD) | P for interaction |
|---|---|---|---|---|
| HOMA-IR | Trp64Trp | 1.49 (1.08) | 1.19 (0.797) | 0.046 |
|  | Trp64Arg or Arg64Arg | 1.68 (1.44) | 1.06 (0.813) |  |
| HOMA-β (%) | Trp64Trp | 67.5 (47.7) | 64.0 (38.2) | 0.479 |
|  | Trp64Arg or Arg64Arg | 68.7 (47.9) | 60.4 (37.8) |  |
| FBS (mg/dL) | Trp64Trp | 98.7 (17.2) | 94.8 (16.8) | 0.192 |
|  | Trp64Arg or Arg64Arg | 101 (18.3) | 93.4 (20.8) |  |
| HbA1c (NGSP) (%) | Trp64Trp | 5.96 (0.526) | 5.88 (0.797) | 0.136 |
|  | Trp64Arg or Arg64Arg | 6.07 (0.771) | 5.84 (0.739) |  |
| Insulin (μIU/mL) | Trp64Trp | 5.99 (3.91) | 5.02 (2.99) | 0.069 |
|  | Trp64Arg or Arg64Arg | 6.50 (4.44) | 4.51 (2.79) |  |

p values for the interaction from a two-way analysis of variance. Hypertension was defined as the use of antihypertensive medication or a blood pressure of 140/90 mmHg or higher. Abbreviations: BP, blood pressure; FBS, fasting blood sugar; HOMA-IR; homeostasis model assessment insulin resistance; HOMA-β, homeostatic model assessment beta cell function; HbA1c (NGSP); hemoglobin A1c (National Glycohemoglobin Standardization Program).

**Table 5. Relationship between HOMA-IR and blood pressure according to the ADRB3 polymorphism at rs4994.**

|  | Model 1<br>odds ratio (95% confidence interval, P value) | Model 2<br>odds ratio (95% confidence interval, P value) | Model 3<br>odds ratio (95% confidence interval, P value) |
|---|---|---|---|
| All subjects | 1.271 (1.038–1.555, 0.020) | 1.305 (1.061–1.606, 0.012) | 1.364 (1.056–1.761, 0.017) |
| Trp64Trp | 1.147 (0.891–1.476, 0.228) | 1.189 (0.918–1.540, 0.191) | 1.215 (0.887–1.664, 0.226) |
| Trp64Arg or Arg64Arg | 1.570 (1.085–2.272, 0.017) | 1.690 (1.133–2.522, 0.010) | 2.054 (1.191–3.541, 0.010) |

The odds ratio (95% confidence interval, P value) was from a multiple logistic regression analysis. Model 1: adjusted for sex, age, and BMI. Model 2: adjusted for sex, age, eGFR, exercise habit, smoking status, drinking habit, and treatment of diabetes. Model 3: adjusted for sex, age, eGFR, exercise habit, smoking status, drinking habit, treatment of diabetes, and HOMA-β. Abbreviations: HOMA-IR; homeostasis model assessment insulin resistance; BMI, body mass index; eGFR, estimated glomerular filtration rate; HOMA-β, homeostatic model assessment beta cell function.

information on the polymorphisms of the adiponectin gene in this study, it is worth investigating in future studies.

The present study had some limitations. Causality was not examined because the study design was cross-sectional. Therefore, further studies are needed to confirm the present results. Furthermore, a selection bias needs to be considered; subjects were voluntary collaborators for the comprehensive health examination, the sample size of which was too small to clarify the effects of the ADRB3 polymorphism. In addition, we did not obtain information on other variables, such as other candidate genes related to insulin resistance and hypertension.

In conclusion, the Trp64Arg polymorphism of ADRB3 was associated with hypertension and insulin resistance in a Japanese population. This relationship, which was dependent on the polymorphism, may predict the development of hypertension and diabetes. The present results need to be interpreted with caution due to the limitations described above.

## Author Contributions

**Conceptualization:** Hiromasa Tsujiguchi, Hiroyuki Nakamura.

**Data curation:** Hiromasa Tsujiguchi, Takayuki Kannon.

**Investigation:** Youhei Yamada, Haruki Nakamura, Hiromasa Tsujiguchi, Akinori Hara.

**Project administration:** Hiroyuki Nakamura.

**Supervision:** Hiromasa Tsujiguchi, Akinori Hara, Sakae Miyagi, Takayuki Kannon, Takehiro Sato, Kazuyoshi Hosomichi, Thao Thi Thu Nguyen, Yasuhiro Kambayashi, Yukari Shimizu, Kim Oanh Pham, Keita Suzuki, Fumihiko Suzuki, Tomoko Kasahara, Hirohito Tsuboi, Atsushi Tajima, Hiroyuki Nakamura.

**Visualization:** Youhei Yamada.

**Writing – original draft:** Youhei Yamada, Haruki Nakamura.

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
