## [Decision Letter · Decision Letter 0]

27 May 2021

PONE-D-20-33381

Relationships among the β3-adrenargic receptor gene Trp64Arg polymorphism, hypertension, and insulin resistance in a Japanese population

PLOS ONE

Dear Dr. Yamada,

Thank you for submitting your manuscript to PLOS ONE. After careful consideration, we feel that it has merit but does not fully meet PLOS ONE’s publication criteria as it currently stands. Therefore, we invite you to submit a revised version of the manuscript that addresses the points raised during the review process.

Your manuscript has been evaluated by one external reviewer and myself (reviewer #2), and the comments are available below.

We look forward to receiving your revised manuscript.

Kind regards,

Raffaella Buzzetti, M.D.

Academic Editor

PLOS ONE

Journal Requirements:

Reviewers' comments:

Reviewer's Responses to Questions

**Comments to the Author**

*Please note that Reviewer #2 is Raffaella Buzzetti, Academic Editor

1. Is the manuscript technically sound, and do the data support the conclusions?

Reviewer #1: Yes

Reviewer #2 : Yes

2. Has the statistical analysis been performed appropriately and rigorously? 

Reviewer #1: No

Reviewer #2: Yes

3. Have the authors made all data underlying the findings in their manuscript fully available?

Reviewer #1: Yes

Reviewer #2: Yes

4. Is the manuscript presented in an intelligible fashion and written in standard English?

Reviewer #1: Yes

Reviewer #2: Yes

5. Review Comments to the Author

Reviewer #1: Major points

1. Could the Authors specify if they used some exclusion criteria to enrol the subjects?

2. It would be interesting if the Authors provide the images of genotyping results.

3. Did The Authors evaluate the sample size and the could they report the power of the study ?

4. All the data were normally distributed? Which kind of test they used to verify data normality?

5. The discussion section is limited, the Authors should add some sentence in which they explain the possible implication of their findings in the clinical practice.

6. The Authors should specific in material and method the range age of subjects enrolled in the study.

7. It would be interesting subdivide the subjects in two groups according gender, to evaluate possible differences with the polymorphisms.

8. In the study there are no information regarding the lipid profile of the subjects. Dyslipidaemia, comprising altered ratio of high TC level and isolated evaluation of the LDL or TG, is usually associated with increased blood pressure (BP) levels. These considerations should be take into account in the discussion section.

Reviewer #2: In this cross-sectional study the Authors evaluated the relationships between Trp64Arg polymorphism of the ADRB3 gene and hypertension and insulin resistance values (HOMA-R) in 719 Japanese subjects. The genotype frequencies of Trp64Trp (homozygous, wild), Trp64Arg (heterozygous, variant), and Arg64Arg (homozygous, variant) were 466 (65%), 233 (32%), and 20 (3%), respectively. Insulin resistance was associated with an increased risk of hypertension in a Japanese population. This relationship was dependent on the presence or absence of the Trp64Arg polymorphism Therefore, the Trp64Arg 47 polymorphism of ADRB3 was associated with hypertension and insulin resistance in this Japanese population.

This is an interesting study, however some major points have to be adressed

1. Have Authors perfomed a power calcualtion before starting the study? Could they report such a calculation?

2. The sentence “In the present study, the ADRB3 polymorphism did not correlate with

phenotypes (obesity, hypertension, and insulin resistance” should be removed from the abstract as it does not add anithing fundamental but,even complicates the comprehension of the text.

3. Please change HOMA-R in HOMA-IR as it is generally reported in the international abbreviation

4. Please report the cut-off for defining Japanese population concerning the BMI ( normal, overweight, obese)

5. Other susceptible genes have been demostrated to be linked to obesity, insulin resistance and hypertension in caucasian population. Among all, adiponectin stands out (PMID: 17030959); please refer to that article and comment on the most important genes already demonstrated to be associated with insulin resistance obesity and hypertension in Japanese population. Surely, it would be very useful for readers

6. Have Authors any informaton about the diabetes type which affected more less 10 percent of the investigated subjects?

7. Exercise and drinking habits should be better quantified. Please comment on that.

6. PLOS authors have the option to publish the peer review history of their article (what does this mean?). If published, this will include your full peer review and any attached files.

Reviewer #1: No

Reviewer #2: No

---

## [Author Response · Author response to Decision Letter 0]

30 Jun 2021

Dear Dr. Raffaella Buzzetti,

Thank you for the thoughtful and constructive feedback you provided regarding our manuscript, ‘Relationships among the β3-adrenargic receptor gene Trp64Arg polymorphism, hypertension, and insulin resistance in a Japanese population’. We have carefully reviewed the comments and revised the manuscript on the basis of the reviewers’ comments. Our point-by-point responses to the reviewers’ comments are listed below this letter. Changes to the manuscript are shown in red font in a separate file labeled 'Revised Manuscript with Track Changes'.

We hope that you find the current version of the manuscript suitable for publication. Thank you for your consideration. We look forward to the publication of our manuscript in the PLOS ONE.

Sincerely,

Youhei Yamada

Reviewer #1

1. Could the Authors specify if they used some exclusion criteria to enrol the subjects?

Response: The participants recruitment chart is shown in the newly created Figure 1.

2. It would be interesting if the Authors provide the images of genotyping results.

Response: We provide the images of genotyping results in the newly created Figure 2.

3. Did The Authors evaluate the sample size and the could they report the power of the study ?

Response: We didn't performed a power calculation before starting the study. After the fact, we confirmed the detection power to see if the sample size was appropriate, but sufficient detection power was confirmed.

4. All the data were normally distributed? Which kind of test they used to verify data normality?

Response: Not all the data were normally distributed. We used Shapiro-Wilk test to verify data normality. In this study, we addressed the problem of normality by using a highly robust analytical method.

5. The discussion section is limited, the Authors should add some sentence in which they explain the possible implication of their findings in the clinical practice.

Response: We increased the text of the discussion section and emphasized the significance of our research results.

6. The Authors should specific in material and method the range age of subjects enrolled in the study.

Response: In the present study, data on 1191 voluntary participants from 40 years of age who underwent the comprehensive health examination between March 2014 and January 2017 were available.

7. It would be interesting subdivide the subjects in two groups according gender, to evaluate possible differences with the polymorphisms.

Response: In response to the proposal, we newly analyzed by gender. The results of this study were found to be prominent in women.

8. In the study there are no information regarding the lipid profile of the subjects. Dyslipidaemia, comprising altered ratio of high TC level and isolated evaluation of the LDL or TG, is usually associated with increased blood pressure (BP) levels. These considerations should be take into account in the discussion section.

Response: We have shown the lipid profile in Tables 1, 2 and 3. Taking into account the lipid profile did not change the trends in our findings.

Reviewer #2

1. Have Authors perfomed a power calcualtion before starting the study? Could they report such a calculation?

Response: We didn't performed a power calculation before starting the study. After the fact, we confirmed the detection power to see if the sample size was appropriate, but sufficient detection power was confirmed.

2. The sentence “In the present study, the ADRB3 polymorphism did not correlate with

phenotypes (obesity, hypertension, and insulin resistance” should be removed from the abstract as it does not add anithing fundamental but,even complicates the comprehension of the text.

Response: I deleted the sentence you pointed out.

3. Please change HOMA-R in HOMA-IR as it is generally reported in the international abbreviation

Response: I changed the word you pointed out.

4. Please report the cut-off for defining Japanese population concerning the BMI ( normal, overweight, obese)

Response: We have shown in Table 1 the cutoff for Japanese BMI.

5. Other susceptible genes have been demostrated to be linked to obesity, insulin resistance and hypertension in caucasian population. Among all, adiponectin stands out (PMID: 17030959); please refer to that article and comment on the most important genes already demonstrated to be associated with insulin resistance obesity and hypertension in Japanese population. Surely, it would be very useful for readers

Response: We mentioned adiponectin in the discussion section with reference to the literature you provided.

6. Have Authors any informaton about the diabetes type which affected more less 10 percent of the investigated subjects?

Response: In our study, type 1 diabetes was excluded from the analysis using HOMAβ.

7. Exercise and drinking habits should be better quantified. Please comment on that.

Response: The lifestyle survey was evaluated qualitatively according to the questionnaire. We are very sorry, but we cannot use quantitative information about exercise and drinking for analysis.

---

## [Editor Report · Decision Letter 1]

19 Jul 2021

Relationships among the β3-adrenargic receptor gene Trp64Arg polymorphism, hypertension, and insulin resistance in a Japanese population

PONE-D-20-33381R1

Dear Dr. Yamada,

We’re pleased to inform you that your manuscript has been judged scientifically suitable for publication and will be formally accepted for publication once it meets all outstanding technical requirements.

Kind regards,

Raffaella Buzzetti, M.D.

Academic Editor

PLOS ONE
---

## [Editor Report · Acceptance letter]

27 Jul 2021

PONE-D-20-33381R1 

Relationships among the β3-adrenargic receptor gene Trp64Arg polymorphism, hypertension, and insulin resistance in a Japanese population 

Dear Dr. Yamada:

I'm pleased to inform you that your manuscript has been deemed suitable for publication in PLOS ONE. Congratulations! Your manuscript is now with our production department. 

Kind regards, 

on behalf of

Dr. Raffaella Buzzetti 

Academic Editor

PLOS ONE